# Individual level analysis of digital proximity tracing for COVID-19 in Belgium highlights major bottlenecks

Caspar Geenen [1,7] ✉, Joren Raymenants [1,2,7], Sarah Gorissen[1], Jonathan Thibaut [1], Jodie McVernon [2,3], Natalie Lorent [4,5] & Emmanuel André[1,6]

To complement labour-intensive conventional contact tracing, digital proximity tracing was implemented widely during the COVID-19 pandemic. However, the privacy-centred design of the dominant Google-Apple exposure notification framework has hindered assessment of its effectiveness. Between October 2021 and January 2022, we systematically collected app use and notification receipt data within a test and trace programme targeting around 50,000 university students in Leuven, Belgium. Due to low success rates in each studied step of the digital notification cascade, only 4.3% of exposed contacts (CI: 2.8-6.1%) received such notifications, resulting in 10 times more cases detected through conventional contact tracing. Moreover, the infection risk of digitally traced contacts (5.0%; CI: 3.0–7.7%) was lower than that of conventionally traced non-app users (9.8%; CI: 8.8-10.7%; $p = 0.002$). Contrary to common perception as near instantaneous, there was a 1.2-day delay (CI: 0.6–2.2) between case PCR result and digital contact notification. These results highlight major limitations of a digital proximity tracing system based on the dominant framework.

Contact tracing aims to slow the spread of an infectious disease. By identifying and alerting contacts (persons exposed to an infectious case or to the same potential source of infection), they can take steps to prevent onward spread, for example by quarantining or testing. Information gathered through contact tracing is also used to study and monitor transmission[1].

Throughout the COVID-19 pandemic, large-scale manual contact tracing (MCT), which involves case interviews to identify contacts, has successfully contributed to limiting disease spread[2]. However, it has well-known weaknesses, such as recall decay and poor scalability. Especially as incidence increased and contact restrictions were eased,

an overwhelmed workforce could result in slower and less comprehensive contact tracing, reducing effectiveness[3,4]. Additionally, central collection of personal identifiable information has caused privacy and security concerns[5,6].

Attempting to mitigate some of these weaknesses, newly developed digital proximity tracing (DPT) through mobile phone apps was implemented in parallel. Using these systems, speed could potentially improve, as contacts are alerted through an automated exposure notification (AEN)[7]. Comprehensiveness could improve, as casual contacts can be alerted of their exposure, even if the index case (the infected person whose contacts are being traced) has no recollection,

[1]KU Leuven, Dept of Microbiology, Immunology and Transplantation, Laboratory of Clinical Microbiology, Leuven, Belgium. [2]Department of Infectious Diseases, The University of Melbourne at the Peter Doherty Institute for Infection and Immunity, Melbourne, Victoria, Australia. [3]Victorian Infectious Diseases Laboratory Epidemiology Unit, Royal Melbourne Hospital at The Peter Doherty Institute for Infection and Immunity, Melbourne, Victoria, Australia. [4]University Hospitals Leuven, Respiratory Diseases, Leuven, Belgium. [5]KU Leuven, Dept of CHROMETA, Laboratory of Thoracic Surgery and Respiratory Diseases (BREATHE), Leuven, Belgium. [6]University Hospitals Leuven, Laboratory Medicine, Leuven, Belgium. [7]These authors contributed equally: Caspar Geenen, Joren Raymenants. ✉e-mail: caspar.geenen@kuleuven.be

personal knowledge or contact details of the exposed person. An automated digital system could also be more scalable than manual case interviews and notifications.

As in MCT, the DPT notification cascade involves a series of steps, many influenced by factors outside the technical workings of the app. When assessing effectiveness, it is useful to analyse each step of the cascade, to identify bottlenecks in the system[8]. First, a proximity event needs to be recorded, requiring both the case and their contact to have installed the app and enabled proximity detection, as well as sufficient technical sensitivity of the detection system (Fig. 1a). When the case later becomes symptomatic (or is identified by other means), a series of time-sensitive steps leads to case diagnosis, contact notification, and eventually altered behaviour (Fig. 1b). The cascade completion rate can be described as the product of the success rates of each step, conditional on having completed the previous one. Therefore, failures in multiple steps can combine to the detriment of DPT effectiveness[9].

In this article, we define the technical sensitivity of a DPT app as the probability that a contact receives an AEN, given that both the case and the contact are active app users and the case triggers notifications. An effective app requires a technical sensitivity sufficient to notify a reasonable proportion of exposed contacts. For the app to be efficient, the proportion of notified contacts who are infected should be high enough to outweigh the societal cost of quarantine or testing. Here, we describe this proportion as the contact's "infection risk" rather than the "secondary attack rate", which seems to imply that the direction of transmission is known[10].

The Google-Apple Exposure Notification framework (GAEN), which directly integrates into the two dominant mobile operating systems (Android and iOS), became the technical backbone of most DPT apps. Despite their promise in modelling studies[3,7,11], doubts were raised from the outset regarding the potential of DPT systems in general and GAEN in particular. First, they require high app uptake, i.e., the proportion of active users in a population. As their efficacy is dependent on both the index case and their contacts being active users, it is proportional to the square of app uptake[7,12,13]. Unfortunately, uptake turned out to be modest at best in most countries, influenced, amongst other factors, by perceived effectiveness, risks to privacy, and trust in science and government[6,14–17]. Second, the limitations of proximity estimation through Bluetooth Low Energy (BLE) signal strength were well known. The types of smartphones, how they are carried, their relative orientation, and the radio environment in which the proximity estimation takes place have a large impact on the estimated exposure risk[18–23].

Experimental field studies of BLE-based exposure notifications—with and without GAEN—registered technical sensitivities under 10% in a healthcare and public transport setting[19,22,24]. Methodological improvements in analytical processing of individual signal strength measurements were deemed, however, to greatly improve accuracy[25,26]. Third, the willingness of anonymously, digitally alerted contacts to follow recommendations may be lower than for manually traced individuals, as has been suggested in several survey studies[27–30] and one cohort study[31]. Fourth, by only recording proximity events rather than location, and storing these locally rather than centralising individual level data, the GAEN system—while safeguarding privacy—limits the study of transmission chains and tracking of certain performance indicators such as the sensitivity, specificity, and timeliness of AENs[32–35].

As a result, empirical evidence on the effectiveness of DPT for COVID-19 is lacking. Some observational studies have used aggregated data gathered by DPT systems to estimate their impact in real life[13,36–38]. Notably, a study on the NHS COVID-19 app in the United Kingdom estimated that it reduced the total number of cases by 5–33% in its first 3 months, with regular use by 28% of the population[39,40]. Although such studies can give an idea of the overall impact of DPT in specific contexts, they cannot compare DPT directly to MCT in terms of overlap in detected cases or timeliness. They have also been unable to quantify technical sensitivity.

These aspects require individual level DPT data, which, as of September 2023, only two previous studies have collected within a real-life contact tracing programme. Vogt et al. evaluated a BLE-based system in New South Wales, Australia[41]. This system, not based on the GAEN framework, stored digitally detected proximity events in a centralised database. During the conventional case interview, contact tracers queried the database for recent digitally registered contacts and determined, along with the index case, the circumstances of their exposure. The proportion of digitally registered contacts fitting the close contact definition for manual contact tracing, i.e., the positive predictive value of DPT for detecting a close contact, was 39%. App-registered contacts who did not fit the criteria were often persons present in the same building, but not the same room. The Zurich SARS-CoV-2 Cohort Study, which evaluated the GAEN-based SwissCovid app by surveying participating cases and contacts, estimated the technical sensitivity at 58%[32]. This study highlighted that only a minority of contacts who were traced both digitally and manually received the AEN before the manual notification[31]. However, it also suggested that non-household contacts who received an AEN quarantined significantly

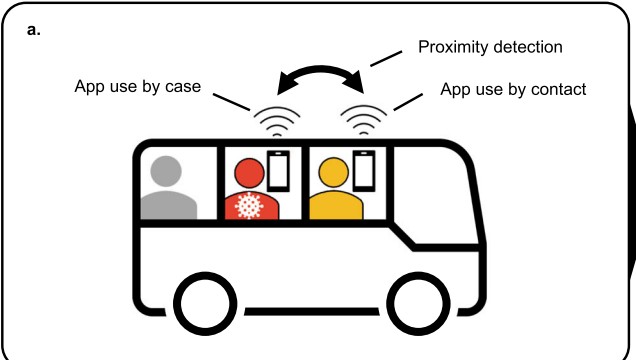

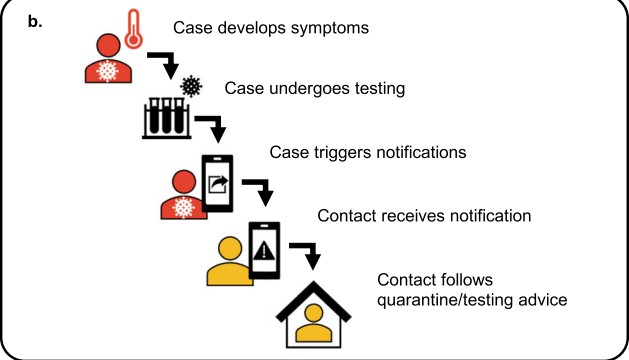

**Fig. 1 | Steps involved in digital proximity tracing (DPT). a** Illustrates three key requirements for DPT to record a proximity event. Both the index case (red) and the contact must have the app installed and use it correctly, implying access to a smartphone, digital and health literacy, and willingness to participate in contact tracing. In this example, one of two casual close contacts on public transport uses the app (yellow), whereas the other (grey) does not. In addition to use of the app, adequate technical sensitivity of the system is required to detect the proximity event. **b** Shows five subsequent time-sensitive steps in the notification cascade: the index case develops a symptomatic infection after the encounter, tests positive, and triggers contact notifications within the app, leading to detected close contacts being alerted (notification receipt) and altered behaviour such as quarantine or testing. Delays or failures in any of these steps would reduce the effect of DPT on epidemic control.

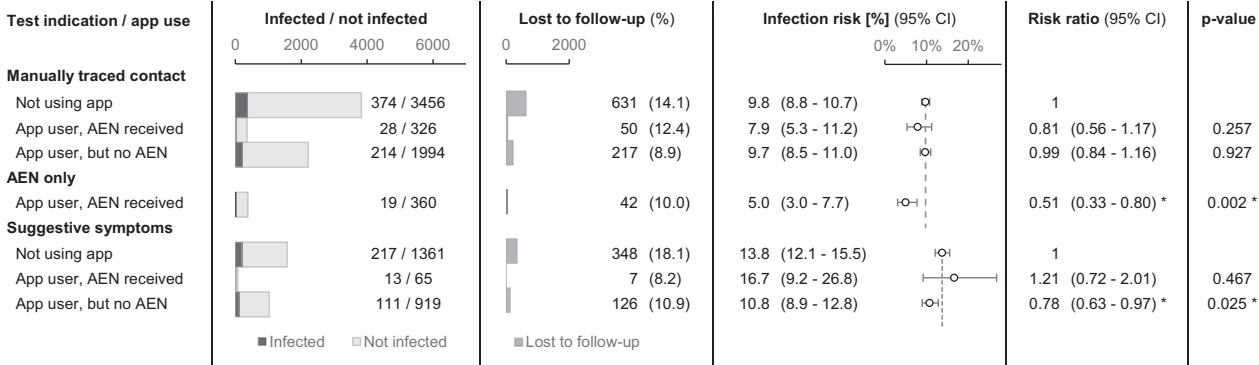

**Fig. 2 | Number of tests, infection risk and risk ratio by self-reported test indication, for persons undergoing a test at the university test centre in the main study period (18 October 2021−9 January 2022).** Persons ($n = 10,878$ invidual test bookings) were grouped according to self-reported main test indication: manually traced close contact, receipt of an automated exposure notification (AEN), or symptoms suggestive for COVID-19. These groups were further subdivided according to app use and AEN receipt. For each subgroup, the number of infected and uninfected persons, the number of persons lost to follow-up, and the infection risk (the proportion subsequently diagnosed with COVID-19) are listed. Error bars indicate the observed infection risks with 95% confidence intervals. In addition, the risk ratio is shown relative to non-app users who were manually traced or had suggestive symptoms. Dashed lines indicate the infection risk in these control groups. Asterisks indicate significant differences using a two-sided Pearson's chi-squared test, not adjusted for multiple comparisons.

faster than those who did not. An important limitation is that, due to low participation rates, selection bias resulted in substantially higher proportions of app uptake and infected app users triggering notifications, compared to both the national and local population[36].

Thus, crucial aspects of the real-life effectiveness of DPT−especially with the GAEN framework−remain understudied, such as the technical sensitivity, the number of cases detected in addition to manual tracing, and the steps in the notification cascade responsible for delays.

In this study, we combined digital and manual contact tracing data on an individual level, to investigate the effectiveness of the Belgian GAEN-based contact tracing app (Coronalert) in a population of higher education students in Leuven. Between October 2021 and January 2022, we systematically questioned cases and their manually traced contacts on their use of the Coronalert app, triggering of notifications, and receipt of an AEN.

Our first aim, relating to the efficiency of DPT, was to quantify the infection risk of students booking an appointment at the university test centre after receiving an AEN. We compared this to a control group of students who attended the test centre solely based on MCT and who denied using the app.

Second, to quantify comprehensiveness and speed of DPT, we sought to determine the proportion of cases and contacts progressing through each step of the notification cascade, and the delays involved in each step.

We combined these results to model the impact of DPT and MCT on the effective reproduction number ($R_{eff}$) in our setting.

## Results

### Infection risk by test indication
Between 18 October 2021 and 9 January 2022, 21,655 PCR test bookings were recorded at the university test centre (exclusion chart: Supplementary Fig. 1). To determine whether these persons were infected, we combined the results of all their tests in the subsequent 14 days. Therefore, any test bookings within 14 days after a previous test (5187; 24.0%) or test booking (1262; 7.7%) were not included as additional observations. In other words, for contacts booking multiple successive tests, only the first reported test indication was considered. Persons were excluded if they already had a positive test in the preceding 60 days (79; 0.5%). From the remaining tests bookings, we excluded another 4219 (27.9%) because the main test indication was not any of the following: suggestive symptoms, an AEN or a manually traced close

contact (see Supplementary Table 1 for accepted test indications). Finally, thirty persons were excluded because of conflicting answers to questions on AEN receipt.

The proportion of these students reporting recent use of the Coronalert app was 41.3% (CI: 40.4–42.2%). The 10,878 included test bookings were divided into three groups according to the main test indication: manually traced close contact (67.0%), suggestive symptoms (29.1%) or an AEN (3.9%). The manual tracing and symptomatic groups were further subdivided according to app use and AEN receipt. The proportion of app users was similar in the manual tracing (38.8%) and symptomatic (39.2%) groups.

In the manually traced group, the overall infection risk was 9.6% (CI: 8.9–10.4%) and not significantly affected by app use or receipt of an AEN (Fig. 2). In the group with suggestive symptoms, the infection risk was significantly lower for app users without an AEN (10.8%; CI: 8.9–12.8%; $p = 0.025$), but not significantly different for app users with an AEN (16.7%; CI: 9.2–26.8%; $p = 0.467$), compared to those not using the app (13.8%; CI: 12.1–15.5%).

The infection risk was 5.0% (CI: 3.0–7.7%) in the group attending only for an AEN, which was significantly lower than for non-app users with a manually traced close contact (9.8%; $p = 0.002$). Similar results were obtained when including all digital and manual notifications: the infection risks differed significantly at 6.4% (CI: 4.7–8.4%) and 9.6% (CI: 8.9–10.4%), respectively ($p = 0.004$).

Using only this data from the test booking forms, we calculated an initial estimate of digital notification cascade completion, from case diagnosis to contact AEN receipt: the proportion of AEN receipt amongst persons attending after being manually traced as a close contact (irrespective of app use or outcome) was 5.5% (CI: 5.0–6.1%). In the next section, we provide a more detailed analysis of the notification cascade and validate this initial estimate.

Similar results were obtained when including all data from the same test centre between 1 February 2021 and 21 March 2022 (Supplementary Figs. 2 and 3). For symptomatic persons tested in this extended study period, we additionally found an increased infection risk in case of a concurrent AEN (risk ratio: 1.34; CI: 1.07–1.67; $p = 0.013$).

### Individual level analysis of the DPT notification cascade
To investigate the comprehensiveness and timeliness of each step in the DPT notification cascade on an individual level, we started from all cases with a positive test in the main study period. We aimed to determine how many of their manually traced close contacts received an AEN, and to identify bottlenecks in the cascade. From the target

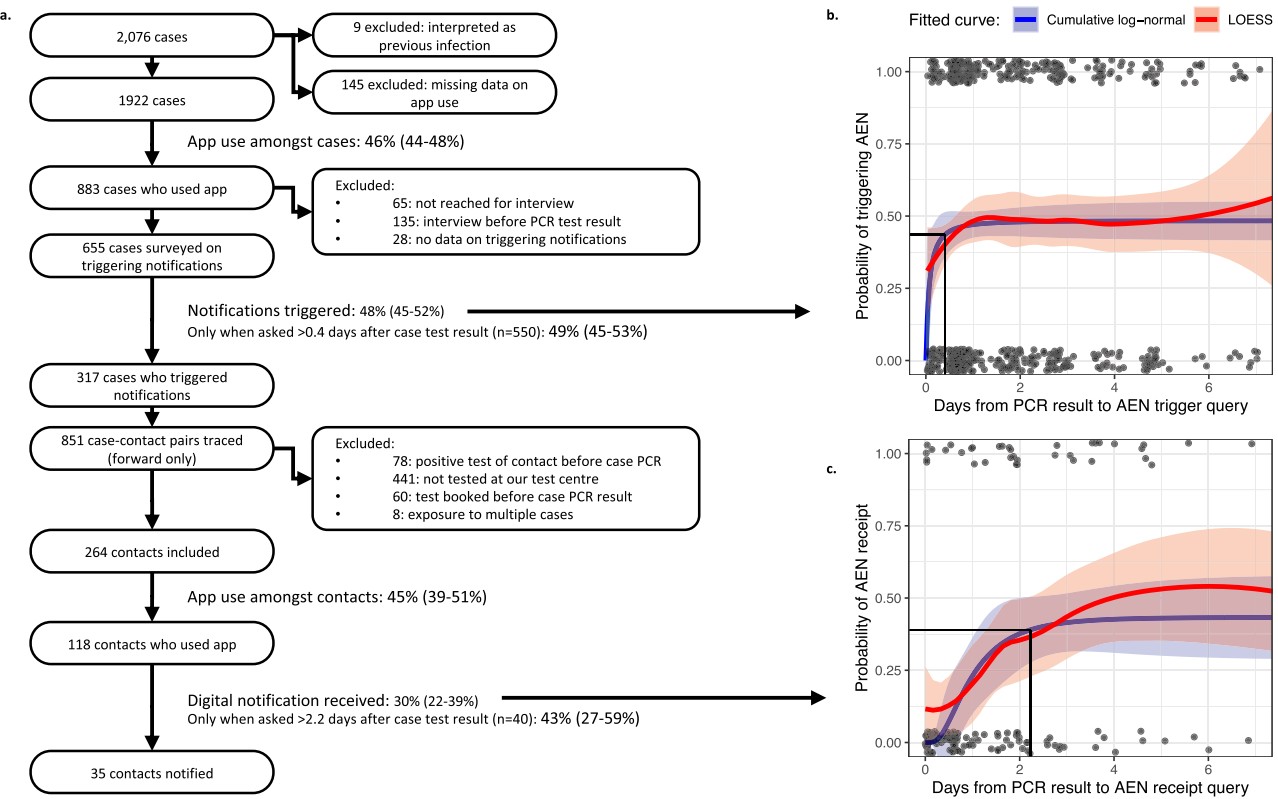

**Fig. 3 | Individual level analysis of the DPT notification cascade. a** Shows the exclusion chart, starting from all cases diagnosed during the main study period (18 October 2021–9 January 2022), and the comprehensiveness of four studied steps in the DPT notification cascade. **b** Shows the proportion of infected app users claiming to have triggered notifications, by time since their PCR result became available (*n* = 550 cases with data for both timepoints). **c** Shows the proportion of contacts indicating that they had received an AEN, by time since confirmation of the PCR result of the index case who triggered notifications. Dots indicate individual data points, with a value of one for success and zero for failure. The red and blue lines show estimates obtained using LOESS and cumulative log-normal regression curves, respectively, with 95% confidence intervals indicated by the shaded areas. For clarity, *x*-axis values over 7 days are not shown, but they are still included for curve fitting. We derive from the cumulative log-normal curves a mean delay of 0.2 (CI: 0.0–0.8) and 1.2 (CI: 0.6–2.2) days, respectively, from case PCR result to their triggering notifications and contact AEN receipt. The time required to reach 90% of the final success rate, and the value of that success rate are indicated with solid black lines.

population, 2,076 cases were reported to the contact tracing team. Nine were excluded because their test result was interpreted as a previous infection or false positive (Fig. 3a).

The proportion of female cases was 54.1% (missing data: 6.4%), while the mean age was 21.6 years (standard deviation: 3.2 years; missing data: 8.1%).

Of the included cases, 46% indicated that they had used the DPT app regularly in the previous week (883 cases; CI: 44%–48%; excluding missing data: 7.0%). App users and non-users had a similar mean age (21.4 and 21.7, respectively) and proportion of female cases (54.9% and 54.5%, respectively). For cases with missing data on sex or age, app use was lower (25% and 26%, respectively).

The university contact tracing team attempted telephone interviews with each of the 883 cases who were app users. The phone call was successful for 818 cases (92.6%), but 135 were excluded because the interview took place before the PCR test result was known (often because of a positive self-administered antigen test, see Methods). Data on whether AENs were triggered was missing for another 28 cases. Of the remaining 655 cases, 48% (317 cases; CI: 45–52%) indicated during their interview that they had authorised the upload of their identifier in the app, triggering contact notifications.

To determine delays in this step, we plotted this proportion by time from PCR test result to case interview, which is when we asked whether notifications had been triggered (Fig. 3b). LOESS regression (locally estimated scatterplot smoothing) suggested that many cases

triggered notifications almost instantly after reporting of the test result. The proportion of cases having triggered notifications then increased, until a plateau was reached <1 day after the PCR result became available. The fitted cumulative log-normal curve indicated a mean delay from case PCR result to triggering of notifications of 0.2 days (CI: 0.0–0.8). By only including cases interviewed after the 90th percentile delay of 0.4 days, the proportion of infected app users eventually triggering notifications was 49% (CI: 45–53%).

Manual forward contact tracing for the 317 cases who confirmed triggering AENs, resulted in 851 case-contact pairs (2.7 contacts per index case), of which 78 were excluded because the contact had already been diagnosed with COVID-19 in the previous 60 days. Out of the resulting pairs, 332 contacts (43%) booked a test at the university test centre, which means they were also students at the same institution. We excluded 60 of these contacts, because their only test booking was before the case PCR result, and 8 because they were already included in a previously identified case-contact pair (they were exposed to multiple cases). Of the remaining 264 contacts, 45% (118 contacts; CI: 39–51%) indicated having used the Coronalert app, similar to the proportion of app users amongst cases.

Assuming that all these contacts were truly exposed to the case, we plotted the answers to the question on whether they had received an AEN, by time elapsed since the case PCR result was confirmed by the laboratory (Fig. 3c). The fitted cumulative log-normal curve showed a mean delay from case PCR result to contact AEN receipt of 1.2 days

**Table 1 | Estimated effectiveness measures of manual and digital contact tracing in our setting**

| | Case isolation with MCT | Case isolation with DPT | Case isolation with MCT and DPT |
|---|---|---|---|
| Notification cascade success rate | 44% (29–63%) | 4.3% (2.8–6.1%) | 47% (30–66%) |
| Mean delay from PCR result to contact notification | 2.3 days (2.1–2.4) | 1.2 days (0.6–2.2) | 2.1 days (2.0–2.4) |
| Mean $R_{eff}$ reduction relative to case isolation only | 158% | 106% | 163% |

Where applicable, 95% confidence intervals are shown between brackets.

(CI: 0.6–2.2 days), with a 90th percentile of 2.2 days. To calculate the technical sensitivity of the Coronalert app, we only included the 40 contacts who answered the question on AEN receipt after the 90th percentile delay.

The technical sensitivity of the app, defined as the proportion of these close contacts who received a notification—given that the case shared the result in the app, both were active users and sufficient time had elapsed—was 43% (17 out of 40 included contacts; CI: 27–59%).

Finally, we combined the success rates obtained for each step of the notification cascade, to estimate the probability that the entire notification cascade (from case diagnosis to contact AEN receipt) was completed for any case-contact pair in our population. Using a simple stochastic model, we estimated this probability at 4.3% (CI: 2.8–6.1%). Basically, this is the result of multiplying the success rate of each step: app use by the case (46%), identifier upload by the case (49%), app use by the contact (45%), and AEN receipt by the contact (43%).

### Estimating the epidemiological impact of manual and digital contact tracing

To estimate the success rate of MCT and combined MCT and DPT in our population, we compared the number of infected contacts who would have been identified by each contact tracing strategy. Based on self-reported test indications, we detected 616 cases through MCT and 341 through symptomatic screening, of which 28 and 13, respectively, also indicated having received an AEN (Fig. 2). In addition, 19 cases were found through DPT alone, without concurrent symptoms or an MCT notification. Thus, a maximum of 60 cases could be identified by DPT, compared to 616 by MCT and 648 by a combined strategy.

If we assume that each notification was triggered by the actual infector, this implies an MCT success rate over 10 times that of DPT. Using the DPT notification success rate of 4.3% determined above, we thus estimated the success rate of the MCT programme at 44% (29–63%), and that of the combined MCT and DPT strategy at 47% (30–66%).

By inputting our estimates of comprehensiveness and speed (Table 1, Supplementary Fig. 4) in a previously published model[42,43], we estimated the effect of different case isolation and contact tracing strategies on the effective reproduction number ($R_{eff}$) in our setting. The effect of each contact tracing strategy on $R_{eff}$ increased almost linearly with the proportion of cases detected through symptomatic screening (Supplementary Fig. 5). Compared to case isolation only, MCT with case isolation achieved the largest reduction in $R_{eff}$ (mean: 1.58 times the effect of case isolation only), while the impact of DPT was minimal in comparison (mean: 1.06 times the effect of case isolation only). Model parameters are listed in Supplementary Table 2. The relative effectiveness of each strategy was robust to variations in input parameters (see Supplementary Table 3).

### Changes in effectiveness over time

Next, we included additional data from the longer period between 1 February 2021 and 21 March 2022, aiming to reveal associations between incidence, engagement with contact tracing, a change in app configuration on 26 April 2021 and the timeliness and comprehensiveness of DPT and MCT (Fig. 4, Supplementary Fig. 6)[44]. There was a downward trend over this period in both app uptake and participation in manual contact tracing. While the number of tests largely followed epidemic waves, DPT contributed only a small minority of test indications throughout, and the fraction of tests performed for symptoms gradually increased with the easing of national test and trace guidelines. The contribution of DPT to test indications did not seem to change substantially with an update to the transmission risk estimation algorithm (aimed at increasing notification thresholds) on 26th April 2021. We saw longer delays in MCT when incidence peaked, indicating a limitation in scalability, but the proportion of DPT as a test indication did not seem to rise with incidence. While the positive predictive value of DPT was usually lower compared to MCT, it rose during high incidence periods, especially the wave attributed to the Omicron variant of concern (VOC) from January 2022.

## Discussion

This study provides unique empirical individual level data on the comprehensiveness and speed of digital proximity tracing for COVID-19. By overlaying the manual and digital contact tracing cascades, we determine bottlenecks in each step of the notification cascade and compare the epidemiological impact of both strategies. As of September 2023, this is the first study to determine the technical sensitivity of a DPT system by consistently querying app use, triggering of notifications and AEN receipt within a manual contact tracing programme. It is also the first to evaluate a DPT system in this later phase of the COVID-19 pandemic, characterised locally by dominance of the Delta and Omicron VOCs, relaxed social contact restrictions, and high vaccination rates.

Previous studies have associated digital notifications with an increased infection risk compared to the general population[29,40]. Additionally in our setting, for app users who were symptomatic, the absence of a concurrent AEN reduced the risk of infection (Fig. 2). In other words, the DPT system provided some information on whether COVID-19 was the cause of any symptoms. For contacts whose exposure had already been established through manual tracing, an AEN did not provide any additional predictive value.

Crucially, for contacts traced only digitally, the infection risk was significantly lower (risk ratio: 0.51, CI: 0.33-0.80) compared to manually traced non-app users, indicating a lower positive predictive value of DPT in the context of this study (Fig. 2). This is consistent with a previous study, which estimated the fraction of digitally notified contacts who fit the close contact definition at only 39% in a centralised BLE-based app[41]. We conclude that the infection risk of digitally traced contacts, although non-negligible (5.0%; CI: 3.0–7.7%), was lower than that of manually traced contacts, indicating less efficient allocation of testing and quarantine.

App uptake was similar amongst cases and contacts at 46% (CI: 44–48%) and 45% (CI: 39–51%), respectively, corresponding with the 48.7% of respondents intending to use the app prior to its launch and the 46% of Belgian smartphone owners who downloaded the app by October 2022[45,46]. This was a high proportion compared to other countries which had well-established DPT apps[13,17,37,40,41].

The proportion of infected app users who triggered notifications (49%; CI: 45–53%) was lower than in early studies on the GAEN-based SwissCovid app (88%) and the NHS COVID-19 app (72%)[31,39]. However, it was similar to later estimates for both apps (37.3–46.3% and 40–55%, respectively)[36,40]. Lower proportions were observed in a study in California (15%) and on a national scale in Belgium (36%)[37,45].

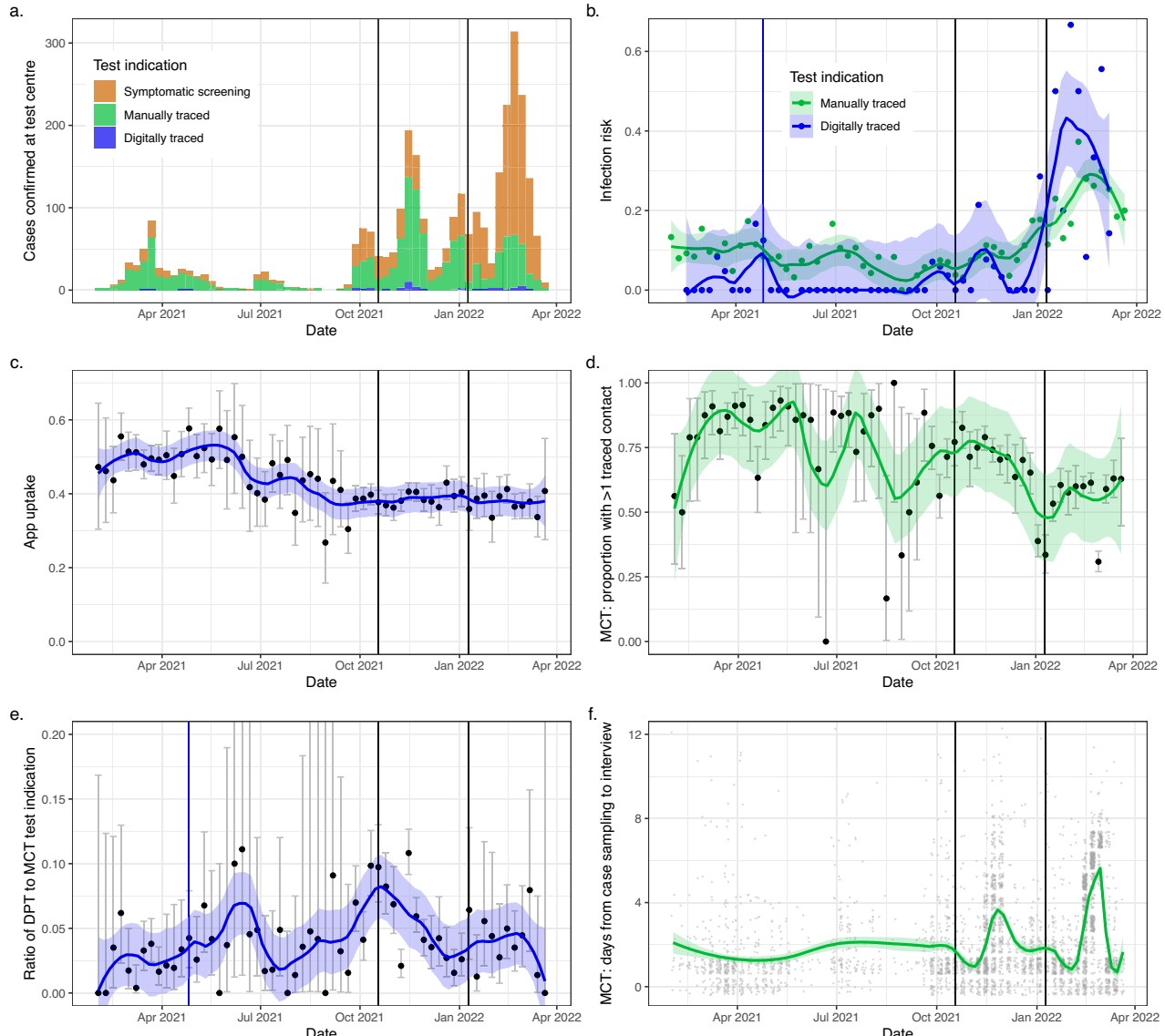

**Fig. 4 | Evolution of test indications and contact tracing performance over time. a, b** Show the number of cases identified and the infection risk (the proportion subsequently diagnosed with COVID-19), respectively, at the university test centre by test indication ($n = 14,907$ MCT/DPT test bookings in **b**). **c, d** Show, respectively, the evolution over time of DPT (digital proximity tracing) app uptake ($n = 24,671$ non-DPT test bookings) and the proportion of cases with at least one contact traced through MCT (manual contact tracing, $n = 6856$ confirmed cases). In (**e**) we show the ratio between persons attending the test centre after a digital and a manual notification ($n = 14,907$ MCT/DPT test bookings). **f** Shows the mean delay

from case PCR sampling to case interview ($n = 4383$ interviewed cases). Dots show weekly observed proportions and ratios, and error bars the 95% confidence interval, except in (**f**) where grey dots are individual data points. Coloured lines show estimates obtained using local polynomial regression curves, and the shaded areas their 95% confidence intervals. Vertical black lines show the beginning and end of the main study period, with the latter corresponding to the end of government-mandated testing for close contacts. A change in app configuration, intended to reduce the number of notified contacts, is indicated with a blue vertical line.

---

We observed a technical sensitivity of 43% (CI: 27–59%) for the Coronalert app, similar to a previous report of 58.5% for the SwissCovid app[32]. This means that, when both the case and the contact used the app and the case triggered notifications, the probability that the contact received a notification was 43%.

Overall, we estimated the probability that the entire DPT notification cascade (from case diagnosis to contact AEN receipt) was completed for any case-contact pair in our population at only 4.3% (CI: 2.8–6.1%), a result of these compounding failure rates throughout the notification cascade (Fig. 5a). This is consistent with the 5.5% (CI: 5.0–6.1%) who had received an AEN, out of all manually traced close contacts attending the test centre, supporting the validity of this latter measure to track DPT comprehensiveness.

Previously reported estimates of app uptake and the proportion of cases triggering notifications are shown in Fig. 5b[13,31,36,37,39–41,47,48]. It is clear that, even in settings with high app uptake, only limited digital cascade completion rates can be achieved, in the absence of efforts to tackle notification sharing and technical sensitivity. Unfortunately, simply changing app parameters to increase technical sensitivity could come at a cost of lowering DPT's positive predictive value, which was already disappointing compared to MCT in this study. If lowering the thresholds for DPT results in a higher quarantine burden per detected case compared to MCT, expanding MCT criteria may be the more efficient intervention.

In terms of timeliness, we observed no significant lag between reporting of a positive PCR result and triggering of AENs by the case,

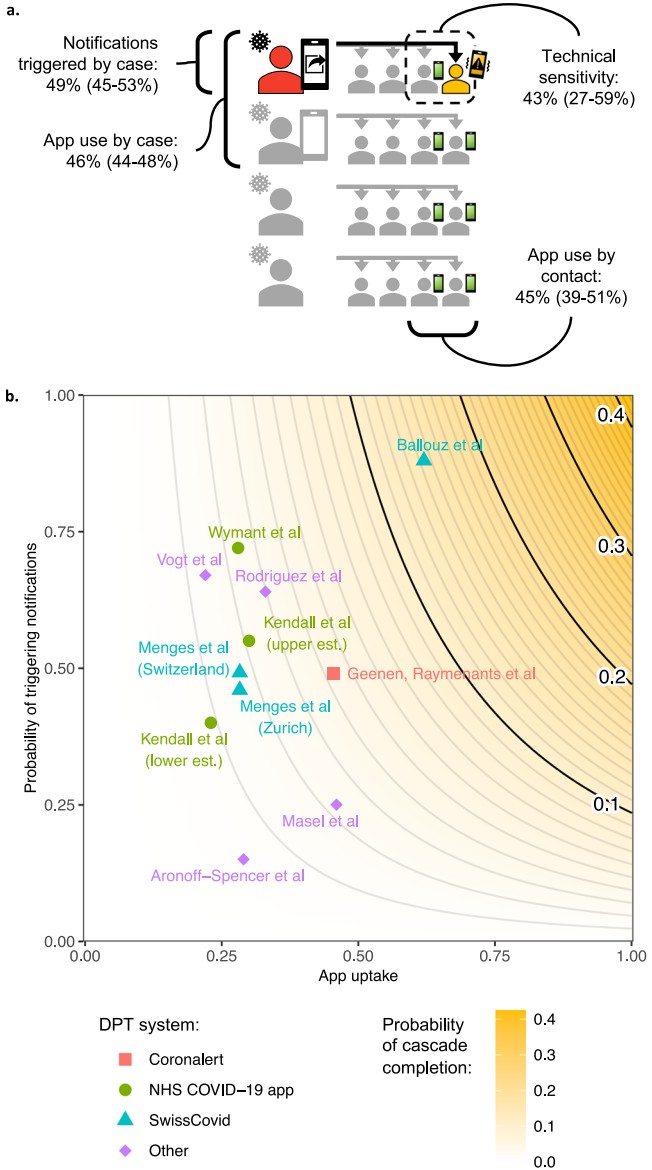

**Fig. 5 | Illustrations of observed success rates in the DPT (digital proximity tracing) notification cascade.** The total probability of completing the notification cascade was estimated at 4.3% (CI: 2.8–6.1). **a** Shows how each step contributed to cascade failures. In a hypothetical scenario of 4 diagnosed cases (left) with 4 contacts each (right), the entire cascade would have been completed for only one out of 16 case-contact pairs (black arrow). Observed success rates are indicated for each step, with 95% confidence intervals between brackets. **b** Compares our results to previous reports of app uptake and the proportion of infected app users triggering notifications, and indicates the expected cascade completion rate when combined with the technical sensitivity we observed for the Coronalert app. Here, we assume that app uptake is equal amongst contacts and cases and that mixing patterns remain as in the studied population. Est.: estimate.

indicating that results were reported rapidly in the app, and index cases consented to DPT promptly, if they intended to do so (Fig. 3b). However, a mean delay of 1.2 days (CI: 0.6–2.2) was observed from case PCR result to contact AEN receipt (Fig. 3c), compared to 2.3 days (CI: 2.1–2.4) for MCT. Possible sources of this delay could lie in the publication of the case identifier on the server, the retrieval and analysis of this information by the contact's device, or the display of an AEN on this device. It is also possible that our questions on DPT as part of this study encouraged cases to trigger notifications after the case interview, leading to delayed notifications. Another explanation might be notifications triggered by a combination of low-risk exposures to

multiple cases. This would mean that an AEN is received only after the last of these cases trigger notifications. Multiple exposures could be especially relevant with regards to superspreading events, or when there is widespread community transmission. Interestingly, Rodriguez et al. observed similar delays in a controlled population experiment on the Spanish Canary Islands, where 98% of index cases who opted to share their code did so within 24 h, while the average delay between a simulated index case introducing the code and the alerted close contact following up with the call centre was 2.35 days[48]. Ballouz et al. also showed that, of contacts who were both manually and digitally notified, only a minority received the AEN before being notified through MCT[31].

In our specific setting, MCT detected over 10 times the number of infected contacts found through DPT alone, although in a combined strategy, DPT could identify 5.2% of additional cases not found through MCT. The resulting modelled effect of MCT on $R_{eff}$ was also nearly 10 times that of DPT alone (58% additional $R_{eff}$ reduction relative to case isolation only, compared to 6%). This was despite a rather low number (mean: 2.7) of manually traced contacts per index case. The model results were robust to variations in input parameters (Supplementary Table 3). As in previous modelling research, the effect of both strategies correlated with the comprehensiveness of detection through diagnostic screening for indications other than contact tracing[43]. This is consistent with the notion of fast, comprehensive detection of (symptomatic) cases as a requirement for effective contact tracing[3].

These results contrast starkly with several early modelling studies estimating DPT effectiveness[3,4,7,12]. While many considered the importance of app uptake, assumptions were likely more optimistic regarding the fraction of index cases triggering notifications, the app's technical sensitivity and the willingness of AEN recipients to follow recommendations to quarantine and undergo diagnostic testing[3,4,7]. Also, they assumed notifications to be instantaneous, which was not true in this study and in several other DPT implementations[31,32,48]. More recent modelling studies highlighted that all exposed contacts, including super spreaders—from which most transmission occurs—are in principle discoverable through manual contact tracing[49]. In contrast, digital proximity tracing can only sample from a limited network of app users, resulting in lower comprehensiveness, which may be more important than speed.

Our effectiveness estimate compared to MCT may appear to contrast with the generally positive assessment in two analyses using aggregated empirical data from the NHS COVID-19 app in the United Kingdom[39,40]. However, based on the estimates of app uptake (23–30%) and the fraction of index cases triggering notifications (40–55%), we would expect a notification cascade success rate similar to our setting (Fig. 5b). Importantly, these studies focused on the absolute epidemiological impact of DPT. They could not make a direct comparison between cases identified by MCT, DPT, symptomatic screening, or a combination, without assumptions regarding the speed, comprehensiveness, and efficiency of different case detection strategies.

In an exploratory analysis, we report additional data from the period of February 2021 to March 2022, aiming to reveal associations between incidence, engagement with contact tracing, a change in app configuration on 26 April 2021 and the timeliness and comprehensiveness of DPT and MCT[44]. The absolute number of individuals undergoing testing after an AEN increased with incidence, as previously reported[36,40]. As an illustration of DPT's scalability advantage, we expected to see an increase in the proportion of traced contacts through DPT relative to MCT during incidence surges. However, we did not observe this sign of scalability, even though the entire DPT cascade could be completed without manual input from a healthcare professional. This could be explained by the equal scalability of the university MCT workforce, consisting mainly of student workers employed on flexible contracts. The change in app configuration on 26 April 2021,

aimed at increasing the threshold for digital notifications, did not noticeably change the fraction of contacts attending the university test centre citing an AEN, relative to manually traced contacts[44]. It is possible that the change did have some effect which was not observed here, due to the small number of observations resulting in broad confidence intervals. Throughout the extended study period, we observed a gradual decline in both app uptake and engagement with MCT (quantified as the fraction of cases reporting at least one contact).

As described above, each of the limitations we observed for the Coronalert app has also been reported for other DPT systems. Thus, the limited evidence suggests common points of failure across different implementations for COVID-19. This does not necessarily indicate intrinsic limitations of DPT. However, some pitfalls might have been avoided if such systems had been developed and refined before the public health emergency, with a view to pandemic preparedness.

For future implementations of DPT, many countries would not resort to mandatory app use to improve uptake, considering the impact on individual rights. More limited mandatory use however, such as the requirement to register attendance at certain high-risk venues, may increase overall uptake, as it did in the United Kingdom[40]. Small financial rewards for using the app may also be considered[6], as well as increasing the general usefulness of the app[16]. To avoid index cases deciding not to share their notification in the app, (monetary) incentives can again be considered. An a priori, once-off consent procedure for exposure notifications in case of a positive test might outweigh any resulting reduction in app uptake or testing[16]. The app's technical sensitivity may benefit from more advanced proximity sensing technologies, such as ultra-wide band (UWB). Extending the contact elicitation window backward in time may help to identify source and sibling cases[50,51]. Finally, the integration of proximity sensing with an estimate of transmission risk in the immediate environment, e.g., through interaction with climate sensors, may further improve the transmission risk estimation algorithm[52].

This study has several limitations. First, our population consisted of highly educated young adults, with near-universal smartphone coverage, which may have resulted in higher-than-average willingness and ability to engage in digital contact tracing[41]. Our young, highly educated population possibly also differed from the general population in other ways, such as the type and number of social interactions, the prior probability of being infected, and the proportion of symptomatic individuals[53,54]. We do not expect these factors to influence our estimates of technical sensitivity which should be independent of app uptake and the proportion consenting to contact tracing. Whereas our observed infection risks could be particularly biased, the significant difference in infection risk between manually and digitally traced contacts suggests a trend generalisable to the broader population.

Second, we could only determine cascade success rates up the point where a contact receives an AEN. Thus, we could not assess whether digitally alerted contacts were less likely to follow quarantine and testing advice. However, our estimate of the epidemiological impact is based on all persons undergoing a test citing a manual or digital notification, thus taking into account any difference in compliance with testing advice. On the other hand, we did not evaluate behavioural changes of digitally alerted app users who did not attend the test centre, which may have contributed to lower onward transmission.

Third, we used interviews and digital questionnaires to obtain information on each step in the digital notification cascade, possibly leading to self-reporting bias[55]. Notably, the number of cases triggering notifications could be overestimated due to social desirability bias.

Fourth, despite a low number of manually traced contacts per case, the majority of cases was detected through MCT, rather than symptomatic screening. This could indicate an unusually effective MCT system or reporting bias, resulting in a distorted comparison with

the epidemiological impact of DPT. However, this should not affect the observed absolute success rate of DPT.

Finally, we could not assess the number of exposures leading to each digital notification. It is possible that many notifications were triggered by a combination of low-risk exposures to multiple cases, rather than a single high-risk exposure. If so, it is possible that DPT could detect more cases than suggested by its ability to notify high-risk contacts. However, our estimate of the epidemiological impact already takes this into account, as it is based on actual numbers of cases detected.

In conclusion, our results confirm that, similar to other DPT apps, the Belgian system was not a replacement for comprehensive MCT. We show that, in a supportive role to MCT, the impact was real but relatively limited. Our individual level analysis of the digital notification cascade highlights limitations in each step, which should be considered in future implementations.

## Methods
### Study type
This observational study complied with the STROBE guidelines.

The study protocol was approved by the Ethics Committee Research UZ/KU Leuven (reference number: S64919). Informed consent was waived by the Ethics Committee, as the data gathered did not exceed what was required for the purpose of safeguarding public health.

### Context
The study was performed in the context of a COVID-19 test and trace programme targeting around 50,000 higher education students at the KU Leuven Association in Leuven, Belgium. This programme was previously described in detail[51,56]. Smartphone coverage was nearly universal, as both internet access and a phone number were required for test booking (in Dutch or English), and we received only a handful of requests for an alternative during the 1.5-year programme. Vaccination rates increased from 2.8% in February 2021 to 10% in May 2021 and over 90% in September 2021[51].

The main study period ran from 18 October 2021—when we started systematically asking cases whether they triggered AENs—to 9 January 2022, when government-mandated testing for all close contacts was abandoned. The Delta and Omicron BA.1 VOCs were dominant[57]. Moderate contact restrictions were in place, with the weighted average Oxford COVID-19 government stringency index for vaccinated and non-vaccinated individuals ranging between 32 and 34[58].

For an additional analysis of variations over time, we also investigated a longer period spanning Alpha, Delta, and Omicron VOC dominance, from 1 February 2021—when an AEN became an accepted test indication at the university test centre—to 21 March 2022, when it stopped accepting contact tracing as a test indication. Contact restrictions were high in the beginning of this period, peaking at an Oxford COVID-19 government stringency index of 76 in April 2021, and declined progressively thereafter, reaching 14 and the end of the period[58].

### Coronalert
The Coronalert mobile phone app was a Belgian government-sponsored implementation of the GAEN framework.

For a notification to be triggered using the GAEN framework, a case needs to undergo a diagnostic test and upload an identifier code to a central database. This upload requires authorisation by the public health authority and explicit permission from the user. The identifier code is then published on a server, retrieved by the contact's device, and contributes to the contact's risk estimation. Once a threshold is reached, an automated notification is displayed to the contact, who can subsequently decide whether to act on it.

Coronalert was released to the public on 30[th] September 2020. It used a simple exposure risk estimator, based on binned Received

Signal Strength Indicator (RSSI) values, timing, and duration of exposure, to determine whether a proximity event with risk of transmission took place. The contact elicitation window started 4 days before symptom onset or positive PCR test of the case, whichever was earlier. The app made a distinction, based on these parameters, between low-risk contacts and high-risk contacts. In this study, we only consider high-risk contacts, because only they received instructions to quarantine and get tested[59].

Parameters used to determine whether a proximity event took place have differed between countries and time periods[26,44,60]. A change to the Coronalert app, intended to reduce the sensitivity of proximity detection, took place on 26 April 2021. Both sets of configuration details can be accessed online[44]. The app was deactivated on 9 November 2022 as the epidemiological situation improved.

Apart from DPT, the Coronalert app provided automatic reports of individual PCR test results and a dashboard with national statistics, such as COVID-19 incidence and vaccination coverage.

Throughout its lifetime, the app was downloaded 4.2 million times, corresponding to 46% of Belgian smartphone owners. No active user numbers were collected on a national level. 1.76 million test results, including 340,000 positive results, were received through the app, which corresponds to 5.4% of the registered COVID-19 tests and 6.9% of the positive COVID-19 tests in the same period. 123,000 persons receiving a positive test through the app proceeded to trigger notifications (36%)[45,61].

As part of the sampling process for COVID-19 tests, healthcare providers in Belgium were instructed to ask app users for a pseudonym code generated by their app. This code was linked to the test in a central database. The app could determine the result of the test by querying a central server for the pseudonym code. App users who had chosen not to input their pseudonym code into the database at the time of sampling, could still do so after receiving their test result, by using an online form or by calling the contact tracing centre.

If the test result was positive, the app automatically prompted the user to consent to notifying their close contacts. If accepted, the identifier code was uploaded to the central database and close contacts received an AEN.

The results of self-administered rapid antigen tests, which gradually become more popular throughout the extended study period, could not be linked to the Coronalert app. All persons with a positive self-administered test were advised to undergo a confirmatory PCR or rapid antigen test, performed by a healthcare professional.

For our analysis, we summarised the notification cascade to four key steps: app use by the index case, identifier upload by the index case, app use by the contact, and receipt of an AEN. We could not determine the proportion of all cases who were diagnosed, or the proportion of notified contacts complying with quarantine or testing advice.

## Data collection

Any student could book a PCR test at the university test centre by filling out an online form, which included the following statements requiring a yes or no answer: "The Coronalert app has been active on my phone for more than a week" and "I received an alert through the Coronalert app that I have had a high-risk contact". They were also asked to input one main test indication, with options including, amongst others, suggestive symptoms, an AEN, and a manually traced close contact. A full list of test indications implemented throughout the study period is added in Supplementary Table 1.

As per national guidelines, Coronalert users attending for a PCR test were consistently encouraged to register their test in the app at the time of sampling, as observed by other researchers conducting in-depth interviews in the same population[16].

A contact tracing team attempted to phone each student with a positive PCR test result for a case interview. They were systematically asked for details of their close contacts and whether they had triggered notifications in the Coronalert app.

For data collection we used Go.Data (version: 2.37.0–build 2105111528), a contact tracing tool developed by the World Health Organisation. Go.Data was integrated with custom-built appointment management, contact listing, and laboratory result modules.

## Study participants: infection risk analysis

In the first part of this study, we determined the infection risk of students attending the test centre, grouped by their test indication and whether they had received an AEN. This analysis compares the positive predictive value of digital and manual notifications.

We included all students who filled out an online test booking form and indicated a manually traced contact, suggestive symptoms, or an AEN as the main test indication. Students were excluded if they had already been identified as infected in the previous 60 days. To avoid selection bias and ambiguous test indications, we also excluded students who had already booked or undergone a test in the preceding 14 days. Finally, we excluded persons who indicated an AEN as the test indication but answered "no" in the same questionnaire, when asked whether an AEN was received.

Students with any positive test in the 14 days after test booking were considered infected, while others with any negative test in the same period were considered not infected. If no test was performed, the student was marked as lost to follow-up.

Risk ratios were calculated relative to control groups of students with the same main test indication who did not use the app.

For students indicating an AEN as the main test indication, we used manually traced non-app users as the control group. As such, we obtain a measure of the positive predictive value of AEN for detecting infection relative to manual contact tracing.

## Study participants: notification cascade

For a detailed analysis of the notification cascade, we started from all infected students referred to the university contact tracing team for a positive test in the study period. This includes students tested at the test centre, but also elsewhere.

During the case interview, we asked whether they had triggered notifications in the Coronalert app. Cases who had used the AEN app were actively encouraged to consent to AEN in their app. They also received basic assistance in case of technical difficulties. If the contact tracer considered the positive test as likely due to a previous infection, or if data on app use was missing, the case was excluded. When determining success rates and delays in the AEN triggering step, we also excluded cases who could not be reached for an interview, cases interviewed only before their PCR test result, and cases with missing information on whether they triggered notifications.

Manual contact tracing was performed for each case, using the same definition for close contact as the national guidelines: either direct physical contact, an interaction at <1.5 m without face masks, an interaction at <1.5 m for >15 min, or an interaction without face masks for >15 min. We additionally labelled as close contacts co-attendants at a "high-risk event" of up to 20 attendees, defined as fitting at least two of the following three criteria: crowding (at least five individuals belonging to at least two households), close contact (within 1.5 m without masks) and closed environment (indoor)[51]. All manually traced close contacts were encouraged to undergo PCR testing as soon as possible and again 7 days after their last exposure, even when this advice differed from national guidelines. After an initial negative PCR test, quarantine until the second test was recommended for all contacts, and a legal requirement for unvaccinated individuals.

Contacts of cases who triggered AENs were included in the analysis if the contact filled out an online PCR test booking form for the test centre, 0 to 14 days after the case PCR result was reported. Contacts were excluded if they already had a positive test up to 60 days

before the case. If a contact was exposed to several cases, only the first reported case-contact pair was retained.

## Analysis of notification cascade

We considered four main steps in the notification cascade: (1) app use by the case, (2) triggering of AEN, (3) app use by the contact, and (4) AEN receipt by the contact. The order of these steps was not chronological but selected to facilitate the analysis. We determined the proportion of cases and contacts progressing through each step, conditional on successful completion of all previous steps.

For the notification trigger step and the AEN receipt step, we plotted the proportion of cases who indicated having triggered notifications, or contacts having received an AEN, by time since PCR test result. For both plots, we fitted the data to a cumulative log-normal curve, assuming that the proportions of cases having triggered AENs and contacts having received an AEN would reach a plateau after a certain delay. The fitted curves were used to estimate the mean delay in each step. We also determined the time required to reach 90% of the supremum, i.e., the time required to reach 90% of the final success rate. Excluding any observations before this time, we determined the proportion of cases and contacts progressing through these two steps of the notification cascade.

To estimate the probability of completing the entire cascade, we used a simple stochastic model. The probability of completing each step, conditional on having completed the previous one, was modelled as a Beta function with shape parameters a (the observed number of successes plus one) and b (the observed number of failures plus one). The probability of completing the entire cascade was modelled by multiplying 100,000 random samples from the probability distributions of each step. A Beta distribution was then fitted to the results, with the mean of the probability density function taken as the estimated probability of completing the cascade. The 95% confidence interval was determined by inputting the values 0.025 and 0.975 in the quantile function of this Beta distribution.

## Effect on epidemic growth

To compare the influence of different case isolation and contact tracing strategies on epidemic growth in our setting, we modelled their effect on the effective reproduction number ($R_{eff}$), which is the average number of secondary infections caused by one case.

The baseline $R_{eff}$ is the result of prevailing general contact restrictions, barrier measures, virological factors (e.g., the dominant VOC) and immunity (natural or vaccine induced) in the absence of case isolation or contact tracing. As the national Belgian $R_{eff}$ varied between 0.69 and 1.68 during the main study period (based on case numbers), we used 1.5 as a baseline[62,63]. We note that, given the transmissibility of Delta or Omicron variants, these values for the baseline $R_{eff}$ can only be achieved with some combination of immunity, general contact restrictions, and barrier measures[64,65].

We modelled the effect of four strategies: case isolation only, case isolation with DPT, case isolation with MCT, and case isolation with DPT and MCT combined.

In the absence of unbiased community prevalence surveys, we varied the comprehensiveness of detection through surveillance (diagnostic screening for indications other than contact tracing) from 0.1 to 0.9 in this model.

To estimate the comprehensiveness of MCT, we first determined the total number of infected contacts identified through MCT and DPT. We assumed that digital or manual exposure notifications were always triggered by the actual infector. We also assumed that our estimate of DPT cascade success rate applied to all contacts attending the test centre, which is possibly incorrect, as this includes contacts exposed to cases outside our target population. We also did not account for persons who did not undergo a test after being manually or digitally notified.

To estimate the speed of case isolation and manual contact notification, we determined the mean delays from symptom onset to test result of symptomatic cases and timing to contact quarantine for their contacts. As we only queried symptoms at the time of the case interview, rather than follow up on symptom development throughout the infection, we could not differentiate pre-symptomatic from asymptomatic cases.

We then inputted these parameters into a previously published compartmental model to determine the effect of different case isolation and contact tracing strategies on $R_{eff}$ (Supplementary Tables 2 and 3)[42,43].

## Extended study period

In an exploratory analysis over an extended time period, with variations in case numbers, vaccination rate, app configuration (a single change to reduce sensitivity took place on April 26 2021), and engagement with MCT and DPT, we plotted the following parameters: confirmed COVID-19 case numbers and infection rate by test indication, app uptake, the percentage of manually traced cases reporting at least one contact (as a measure of active participation in MCT), the delay between PCR test result and MCT (as a measure of speed), and the proportion of contacts identified through DPT relative to MCT.

## Statistical methods

No power analysis was performed, because the study size was a direct result of the number of cases and contacts during the study periods, which were chosen as broadly as possible, as described above. The obtained study size is reflected in the confidence intervals of the results. No randomization or blinding was performed in this observational study.

For continuous variables, t-based two-sided 95% confidence intervals were calculated. For binomial variables, the Clopper-Pearson method was used to determine two-sided 95% confidence intervals, and small sample adjusted risk ratios were determined with two-sided normal -95% confidence intervals. We used a two-sided Pearson's chi-squared test to determine whether there was a difference between two proportions.

For the mean delay of each notification cascade step, we calculated the mean of a (cumulative) log-normal curve fitted to the observed data. The 95% confidence interval was determined as 1.96 times the standard error, either side of the mean on a logarithmic scale. This value was then converted to the equivalent mean on a linear scale. The confidence interval of the digital notification cascade completion rate was determined using a fitted Beta curve, as described above.

When considering outcome data, cases and contacts lost to follow-up were excluded from the analyses.

Data analysis was performed using R scripts specifically written for this study in R version 4.2.2, using the following packages: tidyverse (2.0.0), epitools (0.5-10.1), lubridate (1.9.2), patchwork (1.1.2), DescTools (0.99.48), tti (0.1.0), ggtext (0;1.2), gridExtra (2.3), readxl (1.4.2), ggrepel (0.9.3), fitdistrplus (1.1–8), investr (1.4.2), and MASS (7.3-58.3).

## Reporting summary

Further information on research design is available in the Nature Portfolio Reporting Summary linked to this article.

# Data availability

The individual-level data underlying the analyses in this study are provided in the Supplementary Data file. The raw age data are available under restricted access for privacy reasons. Access to age data can be obtained by request to the corresponding author (C.G.), who will respond within 4 weeks. There must be a demonstrable affiliation with an academic or health institution, a legitimate epidemiological question, and a commitment not to attempt to de-anonymise.

## Code availability

Code to reproduce these results is available from: https://github.com/c-geenen/DPT-leuven[66].

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

## Acknowledgements

We thank Bart Preneel for helpful comments and suggestions. We thank Klaas Nelissen for co-designing the university contact tracing programme and the associated software implementation. The test centre was funded by both the National Institute for Health and Disability Insurance (RIZIV/INAMI) and the regional Flemish government's Agentschap voor Zorg & Gezondheid. J.R. and J.T. acknowledge support of the Research Foundation Flanders (FWO, grant numbers: 1S88721N to J.R.; 1130423 N to J.T.). This study was financed through internal KU Leuven funds and the DURABLE project. The DURABLE project has been co-funded by the European Union, under the EU4Health Programme (EU4H), Project No. 101102733. Funded by the European Union. Views and opinions expressed are however those of the authors only and do not necessarily reflect those of the European Union or the European Health and Digital Executive Agency. Neither the European Union nor the granting authority can be held responsible for them.

## Author contributions

C.G., J.R. and E.A. conceptualized the study. C.G. and J.R. designed the analysis. C.G., J.R., and J.T. performed the analysis. C.G. and J.R. wrote the manuscript. All authors (C.G., J.R., S.G., J.T., J.McV., N.L. and E.A.) critically revised the manuscript.

## Competing interests

The authors declare no competing interests.
