## [Peer Review File · Nature Communications]

Individual level analysis of digital proximity tracing for COVID-19 in Belgium highlights major bottlenecksREVIEWER COMMENTS

Reviewer #1 (Remarks to the Author):

I wholeheartedly complement the authors with their paper. The amount of work required and breadth of the analyses undertaken with this empirical dataset, including a model-based extension, are truly to be commended.

I have a few comments for your consideration, but in general the paper is complete and methodologically sound in my opinion.

Comments

- Abstract:

- o Could you mention the sample size of the population studied?

- Introduction:

- o You mention observational studies, such as the UK NHS tracing app. I would recommend distinguishing between model-based studies (of which there are a lot), empirical studies (of which there are few), and combinations of these. For an overview of these Effectiveness of contact tracing apps for... | F1000Research

- o In the subsequent paragraph, please update this statement (These aspects require individual-level DPT data, which, as of July 2023, only two previous studies have collected within a real-life contact tracing programme) with the other empirical studies from the review above

- Methods:

- o No comments

- Results:

- o No comments (there are multiple typos for the word notifications in the cascade figure 3)

- Discussion:

- o I would recommend mentioning the empirical data used in this study. I really consider this a strength, as there are very few empirical studies (and many many model-based ones). So for example: unique empirical individual level data

- o Could you comment on why it is the case that the number of triggered notifications is lower in your study compared to other studies? (UK, Swiss)

- o Your limitation of highly educated app users is more broad. Not only is it about willingness to engage in digital contact tracing (adoption rate), this sample might also differ from the general population in many other factors. There likelihood of getting infected (so the prior probability), the interactions with others (compared to for example elderly who are homeconfined more), the fraction of asymptomatic individuals, etc. Could you elaborate more on this, including relevant references describing the difference between your sample (young highly educated) to the general population

Reviewer #2 (Remarks to the Author):

This is an interesting manuscript on the limitations of Digital Contact Tracing (DCT) or Digital Proximity Tracing (DPT) app implementations.

I really appreciate the focus on the "funnel dropoff" of app effectiveness. This issue was raised early in the pandemic (see e.g. this blog <https://tomaspuoyo.medium.com/coronavirus-learning-how-to-dance-b8420170203e>) but there has been little follow-up, especially from the scientific community. There has been another excellent preprint on the same topic of multiple points of failure for DCT by Masel et al (<https://arxiv.org/pdf/2306.00873.pdf>) that should absolutely be cited and discussed here in this context.

I also appreciate the focus on a relatively successful (at least in terms of uptake) yet ineffective

DCT implementation.

Having contributed to the development of another EU GAEN-based app that was unfortunately a failure from a public health perspective, I think it is important to think carefully about the points of failure and learn what went wrong with different deployments.

The results of this ms are interesting and clear. Since there are 4 points of failure that could be detected in the context of this Leuven study on the Belgian app, and all these 4 points show a success rate around 40-50%, the app effectiveness can be summarised as $(40-50\%)^4 \sim 5\%$. This is a beautiful illustration of the whole concept of cascade of points of failure.

There are a few issues with this paper, which concern mostly the presentation and discussion of the findings rather than the results per se.

i) there are many places in the ms where the authors seem to claim a generality for their results that is not warranted, referring to the limitations of DCT instead of the specific app/intervention (and related parameters chosen by the Belgian health authorities). Related to this, there is a lack of clarity on the intrinsic vs contingent nature of these limiting factors.

For example, the fact that only a fraction of users decided to trigger an AEN is a strong limitation, but it's not actually intrinsic to DCT or any general implementation; it's simply a quirk imposed by Google and Apple as part of their conditions for GAEN.

The delays between test results and AENs are also extremely dependent on the implementation, and could be technically compressed to less than an hour. Lines like 395-396 are pointless, since there is no doubt that DPT can technically be practically instantaneous.

The Belgian app uptake has been relatively good during the pandemic, but it is well known that higher uptakes are needed for an effective response, and they are possible in principle (but probably not with top-down opt-in incentive-free approaches). The authors themselves discuss some strategies to increase uptake in the Discussion.

In my opinion, in order for this ms to be a more productive contribution to the scientific debate, the authors should clarify which factors are contingent to the Belgian implementation and which ones represent more general issues with DCT.

ii) A major limitation of this study is that it assumes effective MCT.

But MCT was apparently as disappointing in Belgium as in most western countries (detecting 2.7 contacts per index case, an order of magnitude less than the expected range). How does this affect their results? The initial analysis (lines 153-156 vs 181-185) makes me think that it should have a minor effect, but the authors should clarify.

Authors should also describe more clearly parameters and effectiveness of manual tracing in Belgium and within the university, since it is used as benchmark.

Related to this: In lines 321-327, the overall SAR for MCT is actually 9.6%, and the overall SAR for DPT is 6.4%. I think this is the most meaningful comparison between the two approaches, rather than the one presented in the text that ignores overlapping MCT/DPT notifications.

The SAR is not an issue in itself, as the authors acknowledge in lines 363-368, since it depends on the app setting&policy (and on MCT policy). What actually matters is the combination of SAR and number of AENs per case using the app. This seems to be a surprisingly low number, i.e. ~ 0.1 declared notifications per case (35/317), if computed directly, or ~ 0.5 ($0.43 \cdot 0.45 \cdot 2.7$) if computed indirectly. These values are really low. Is it possible that most of the app notifications are actually not included in this study?

iv) The long delay from PCR results to AEN received is a surprising result. Lines 369-380 outline that the delay has little to do with DCT in itself. It originates instead from implementation issues that were probably due to the rapid deployment. Lines 514-516 may represent a plausible explanation for the delay, if most app users would not give their pseudonym code at the time of sampling. There are also a few surprising findings about the (in)effectiveness of the Belgian app. I do not doubt the analysis, but all together, they paint a curious picture. For example, lines 414-416 report how there was no hint of DPT scalability in Belgium. Furthermore, lines 416-419 and figure 4e report how there was no change in the relative number of notifications with a change in AEN threshold. Something weird is going on here with the performances of the Belgian app.

Taken together, these findings outline the limitations inherent in new approaches developed and

deployed during an epidemic, and emphasise the role of pandemic preparedness. The authors should shift their emphasis towards this aspect, which is already mentioned in the ms.

v) The app seems to have very poor engagement overall in Belgium, at least judging from actual tests received through the app (around 5-10% of tests, despite a potential user base of 46% of the Belgian population, lines 505-510). Hence, either students were anomalously well engaged with the app, or I wonder how many of them lied when questioned about having triggered AENs or not. I believe the authors can find a good bunch of papers in behavioural science about the odds of admitting a socially undesirable behaviour during interviews.

vi) Curiously, the risk among manually traced contacts is very similar to the risk for individuals showing symptoms (10 vs 14%), despite a large peak of Delta cases and relatively low levels of influenza-like illnesses in that period would suggest a high specificity for symptom-based detection.

The finding that infection risk was lower for app users without an AEN than non-app users would also suggest that the AEN was more informative in the presence of symptoms than without. I think there is some serious issue here, possibly regarding the declared reason for testing. Was there any difference in the legal implications of different reasons given for seeking a test? I understand that the whole issue may be hard to investigate.

Minor points:

lines 76-77: I can understand the issue of the authors with the term "secondary attack rate", but this term is vaguely defined in the literature and used quite often in the same context as "infection risk" in this ms, even when the direction of transmission is not known.

Lines 96-98: "cannot be used" is not technically correct and it is far too strong. It is true that GAEN poses significant limits.

Line 103: to my memory, there is at least one study cited there (Wymant et al) that infers the impact of DCT based on statistical regression rather than modelling (and therefore does not require especially strong assumptions).

Point-by-point response to reviewer comments

We would like to thank both reviewers for their insightful comments and suggestions. We believe that the amendments detailed below fully address their concerns.

Reviewer #1 (Remarks to the Author)	Point-by-point response
I wholeheartedly complement the authors with their paper. The amount of work required and breadth of the analyses undertaken with this empirical dataset, including a model-based extension, are truly to be commended. I have a few comments for your consideration, but in general the paper is complete and methodologically sound in my opinion.	N/A
Abstract:  • Could you mention the sample size of the population studied? 	The approximate size of the target population (50,000) was added to the abstract.
Introduction:  • You mention observational studies, such as the UK NHS tracing app. I would recommend distinguishing between model-based studies (of which there are a lot), empirical studies (of which there are few), and combinations of these. For an overview of these Effectiveness of contact tracing apps for... F1000Research 	We appreciate this comment, and have made a few changes to clarify the types of studies described:  - Modelling studies: “Despite their promise in modelling studies, [...]” (line 70) - Observational studies with aggregated data (including those with a modelling component): “As a result, empirical evidence on DPT is lacking. Some observational studies have used aggregated data gathered by DPT systems to estimate their impact in real life.” (lines 90-92) - Observational studies with individual-level data: “These aspects require individual level DPT data, which, as of September 2023, only two previous studies have collected within a real-life contact tracing programme.” (lines 97-98)
 • In the subsequent paragraph, please update this statement (These aspects require individual-level DPT data, which, as of July 2023, only two previous studies have collected within a real-life contact tracing programme) with the other empirical studies from the review above 	Thank you for this suggestion. We identified two empirical studies from this review:  - Kendall et al used aggregated data to estimate the impact of DPT on the Isle of Wight. We have added a reference to this study in the paragraph on observational studies using aggregated data (line 92). - Chen et al investigated a Public Warning System, used to contain an outbreak related to the Diamond Princess cruise ship. An alert was sent via SMS, advising the population to quarantine if they had visited any of 39 specific locations. We decided not to reference this study here, because the Public Warning System was very different from DPT as described in our manuscript.
Methods:  • No comments 	N/A
Results:  • No comments (there are multiple typos for the word notifications in the cascade figure 3) 	Typos corrected in Figure 3.
Discussion:  • I would recommend mentioning the empirical data used in this study. I really consider this a strength, as there are very few empirical studies 	Revised as suggested (line 260).

(and many many model-based ones). So for example: unique empirical individual level data	
 • Could you comment on why it is the case that the number of triggered notifications is lower in your study compared to other studies? (UK, Swiss) 	It seems that the proportion triggering notifications was higher in the start-up phase of DPT apps in the UK and Switzerland. We clarified the comparison of this metric to other studies as follows: “The proportion of infected app users who triggered notifications (49%; CI: 45-53%) was lower than in early studies on the GAEN-based SwissCovid app (88%) and the NHS COVID-19 app (72%). However, it was similar to later estimates for both apps (37.3-46.3% and 40-55%, respectively).” (lines 287-289) Related to this, we removed a possibly misleading datapoint in Figure 5. The figure showed our own back-of-the-envelope estimate of 69.9% for the Swiss app, based on an article which reported the total number of confirmed cases in Switzerland (12,456), the number of cases triggering notifications in the app (1,645), and the proportion of active users in the population (18.9%), resulting in an estimated $1645/(12456*0.189) = 69.9\%$ of infected app users triggering notifications (https://doi.org/10.4414/smw.2020.20457). We have removed this questionable datapoint, since we also reference peer-reviewed estimates for the Swiss app.
 • Your limitation of highly educated app users is more broad. Not only is it about willingness to engage in digital contact tracing (adoption rate), this sample might also differ from the general population in many other factors. There likelihood of getting infected (so the prior probability), the interactions with others (compared to for example elderly who are homeconfined more), the fraction of asymptomatic individuals, etc. Could you elaborate more on this, including relevant references describing the difference between your sample (young highly educated) to the general population 	We have elaborated further on this limitation in Discussion: “Our young, highly educated population possibly also differed from the general population in other ways, such as the type and number of social interactions, the prior probability of being infected, and the proportion of symptomatic individuals. We do not expect these factors to influence our estimates of technical sensitivity, which should be independent of app uptake and the proportion consenting to contact tracing. Whereas our observed infection risks could be particularly biased, the significant difference in infection risk between manually and digitally traced contacts suggests a trend generalisable to the broader population.” (lines 393-400)

Reviewer #2 (Remarks to the Author)	Point-by-point response
This is an interesting manuscript on the limitations of Digital Contact Tracing (DCT) or Digital Proximity Tracing (DPT) app implementations.	N/A
I really appreciate the focus on the "funnel dropoff" of app effectiveness. This issue was raised early in the pandemic (see e.g. this blog https://tomaspuoyo.medium.com/coronavirus-learning-how-to-dance-b8420170203e) but there has been little follow-up, especially from the scientific community. There has been another excellent preprint on the same topic of multiple points of failure for DCT by Masel et al (https://arxiv.org/pdf/2306.00873.pdf) that should absolutely be cited and discussed here in this context.	As suggested, the preprint by Masel et al was referenced in Introduction, with a short additional description: “The cascade completion rate can be described as the product of the success rates of each step, conditional on having completed the previous one. Therefore, failures in multiple steps can combine to the detriment of DPT effectiveness.” (lines 58-60)

I also appreciate the focus on a relatively successful (at least in terms of uptake) yet ineffective DCT implementation. Having contributed to the development of another EU GAEN-based app that was unfortunately a failure from a public health perspective, I think it is important to think carefully about the points of failure and learn what went wrong with different deployments.	N/A
The results of this ms are interesting and clear. Since there are 4 points of failure that could be detected in the context of this Leuven study on the Belgian app, and all these 4 points show a success rate around 40-50%, the app effectiveness can be summarised as $(40-50\%)^4 \sim 5\%$. This is a beautiful illustration of the whole concept of cascade of points of failure.	N/A
There are a few issues with this paper, which concern mostly the presentation and discussion of the findings rather than the results per se.	N/A
i) there are many places in the ms where the authors seem to claim a generality for their results that is not warranted, referring to the limitations of DCT instead of the specific app/intervention (and related parameters chosen by the Belgian health authorities). Related to this, there is a lack of clarity on the intrinsic vs contingent nature of these limiting factors. For example, the fact that only a fraction of users decided to trigger an AEN is a strong limitation, but it's not actually intrinsic to DCT or any general implementation; it's simply a quirk imposed by Google and Apple as part of their conditions for GAEN. The delays between test results and AENs are also extremely dependent on the implementation, and could be technically compressed to less than an hour. Lines like 395-396 are pointless, since there is no doubt that DPT can technically be practically instantaneous. The Belgian app uptake has been relatively good during the pandemic, but it is well known that higher uptakes are needed for an effective response, and they are possible in principle (but probably not with top-down opt-in incentive-free approaches). The authors themselves discuss some strategies to increase uptake in the Discussion. In my opinion, in order for this ms to be a more productive contribution to the scientific debate, the authors should clarify which factors are contingent to the Belgian implementation and which ones represent more general issues with DCT.	This is indeed an important point. To clarify, we do not claim that any limitations are directly generalisable to other DPT systems. We also do not make any claims on intrinsic limitations of DPT, because our results do not provide limits on what could be achieved in an ideal scenario. However, we would like to make the point that each of the observed limitations (technical sensitivity, delays, app uptake, notification triggering, and positive predictive value) has also been observed in other DPT implementations for COVID-19, suggesting common points of failure. This was highlighted in an additional Discussion paragraph (lines 373-377). We have also emphasised the contingent nature of results and limitations as follows:  • Abstract: “These results highlight major limitations of a digital proximity tracing system based on the dominant framework.” • Lines 270-271: “Additionally in our setting, for app users who were symptomatic, the absence of a concurrent AEN reduced the risk of infection.” • Line 292: “We observed a technical sensitivity of 43% (CI: 27-59%) for the Coronalert app [...]” • Lines 340-344: “[In early modelling studies] assumptions were likely more optimistic regarding the fraction of index cases triggering notifications, the app’s technical sensitivity and the willingness of AEN recipients [...]. Also, they assumed notifications to be instantaneous, which was not true in this study and in several other DPT implementations.” • Lines 349-350: “Our effectiveness estimate compared to MCT may appear to contrast with the generally positive assessment in two analyses using aggregated empirical data [...]” • Lines 417-418: “In conclusion, our results confirm that, similar to other DPT apps, the Belgian system was not a replacement for comprehensive MCT.”

	Although the delay in receiving a notification is indeed dependent on implementation, there is at least one type of situation where compressing it to near-instantaneous would not be possible in practice. Notifications caused by combined low-risk exposures to multiple cases can only be received after the last case triggers notifications. This scenario could be quite common, for example due to superspreading events or when there is widespread community transmission. We have explained this in the manuscript (lines 321-325).
ii) A major limitation of this study is that it assumes effective MCT. But MCT was apparently as disappointing in Belgium as in most western countries (detecting 2.7 contacts per index case, an order of magnitude less than the expected range). How does this affect their results? The initial analysis (lines 153-156 vs 181-185) makes me think that it should have a minor effect, but the authors should clarify.	Comprehensiveness of MCT is indeed difficult to quantify in the absence of unbiased incidence data. Therefore, we do not make any assumptions on MCT comprehensiveness, except when modelling the effects of MCT and DPT on the effective reproduction number in our setting. We have emphasised that the comparison applies only to our specific setting (line 329) and that the number of manually traced contacts per index case was relatively low (lines 332-333). In the analysis of the notification cascade, we do assume that all manually traced contacts were truly exposed (clarified, line 207).
Authors should also describe more clearly parameters and effectiveness of manual tracing in Belgium and within the university, since it is used as benchmark.	The close contact definition for manual contact tracing is detailed on line 535-545, together with the recommendations regarding testing and quarantine. Several indicators of MCT performance are shown in Figure 4, panels d/f and Supplementary Figure 6, panel d.
Related to this: In lines 321-327, the overall SAR for MCT is actually 9.6%, and the overall SAR for DPT is 6.4%. I think this is the most meaningful comparison between the two approaches, rather than the one presented in the text that ignores overlapping MCT/DPT notifications.	The two groups were chosen to reduce the risk of bias as much as possible. However, as suggested, we also calculated alternative infection risks, including overlapping MCT/DPT notifications: “Similar results were obtained when including all digital and manual notifications: the infection risks differed significantly at 6.4% (CI: 4.7-8.4%) and 9.6% (CI: 8.9-10.4%), respectively (p=0.004).” (lines 157-158)
The SAR is not an issue in itself, as the authors acknowledge in lines 363-368, since it depends on the app setting&policy (and on MCT policy). What actually matters is the combination of SAR and number of AENs per case using the app. This seems to be a surprisingly low number, i.e. ~0.1 declared notifications per case (35/317), if computed directly, or ~0.5 (0.43*0.45*2.7) if computed indirectly. These values are really low. Is it possible that most of the app notifications are actually not included in this study?	We could obtain a rough estimate of the number of notifications per case in two ways:  • Dividing the number of individuals attending the test centre citing an AEN (with or without other test indications: 910 individuals) by the number of cases who confirmed triggering notifications (317), results in 2.9 notifications per case triggering them. However, this number does not consider any cases or contacts who were excluded, any contacts who did not undergo testing, or any exposures outside the target population. • An indirect calculation would require knowledge of the true number of exposed individuals per case. For example, the true number of contacts per case might be 10. Multiplying this with contact app uptake and technical sensitivity results in $10 * 0.45 * 0.43 = 1.9$ notifications per case triggering them. Therefore, we do not claim to have sufficient information to accurately calculate this metric. However, the rough estimates seem comparable to other reports (e.g., https://doi.org/10.1038%2Fsa41467-023-36495-z).

iv) The long delay from PCR results to AEN received is a surprising result. Lines 369-380 outline that the delay has little to do with DCT in itself. It originates instead from implementation issues that were probably due to the rapid deployment. Lines 514-516 may represent a plausible explanation for the delay, if most app users would not give their pseudonym code at the time of sampling. There are also a few surprising findings about the (in)effectiveness of the Belgian app. I do not doubt the analysis, but all together, they paint a curious picture. For example, lines 414-416 report how there was no hint of DPT scalability in Belgium. Furthermore, lines 416-419 and figure 4e report how there was no change in the relative number of notifications with a change in AEN threshold. Something weird is going on here with the performances of the Belgian app. Taken together, these findings outline the limitations inherent in new approaches developed and deployed during an epidemic, and emphasise the role of pandemic preparedness. The authors should shift their emphasis towards this aspect, which is already mentioned in the ms.	As suggested, we have further emphasised the importance of pandemic preparedness: “However, some pitfalls might have been avoided if such systems had been developed and refined before the public health emergency, with a view to pandemic preparedness.” (lines 376-377) As we mention above (i) we disagree that accurate, instantaneous digital notifications are by definition achievable with BLE technology, a point which we clarified in the text (lines 319-323). Lines 514-516 (now lines 478-480) describe how the test result was linked to the app, which could be done at the time of testing or later. However, as established in Results, we observed no considerable delay between the PCR result being confirmed by the laboratory (irrespective of whether it was reported in the app) and the case triggering notifications. Thus, we discount this as an explanation for the observed delays in AEN receipt by contacts. We have provided two additional explanations for the observed delays: “It is also possible that our questions on DPT as part of this study encouraged cases to trigger notifications after the case interview, leading to delayed notifications. Another explanation might be notifications triggered by a combination of low-risk exposures to multiple cases. This would mean that an AEN is received only after the last of these cases triggers notifications. Multiple exposures could be especially relevant with regards to superspreading events, or when there is widespread community transmission.” (lines 317-323). We have provided a possible explanation for not observing scalability of DPT relative to MCT: “This could be explained by the equal scalability of the university MCT workforce, consisting mainly of student workers employed on flexible contracts.” (lines 367-369) We also added a plausible explanation for not observing an effect of the change in app parameters: “It is possible that the change did have some effect which was not observed here, due to the small number of observations resulting in broad confidence intervals.” (lines 369-370)
v) The app seems to have very poor engagement overall in Belgium, at least judging from actual tests received through the app (around 5-10% of tests, despite a potential user base of 46% of the Belgian population, lines 505-510). Hence, either students were anomalously well engaged with the app, or I wonder how many of them lied when questioned about having triggered AENs or not. I believe the authors can find a good bunch of papers in behavioural science about the odds of admitting a socially undesirable behaviours during interviews.	We have mentioned the possibility of self-reporting bias as an additional limitation: “Third, we used interviews and digital questionnaires to obtain information on each step in the digital notification cascade, possibly leading to self-reporting bias. Notably, the number of cases triggering notifications could be overestimated due to social desirability bias.” (lines 408-410).
vi) Curiously, the risk among manually traced contacts is very similar to the risk for individuals showing symptoms (10 vs 14%), despite a large peak of Delta	The university test centre made efforts to lower thresholds for testing: tests were free of charge and were encouraged in targeted advertising even for symptoms of

cases and relatively low levels of influenza-like illnesses in that period would suggest a high specificity for symptom-based detection. The finding that infection risk was lower for app users without an AEN than non-app users would also suggest that the AEN was more informative in the presence of symptoms than without. I think there is some serious issue here, possibly regarding the declared reason for testing. Was there any difference in the legal implications of different reasons given for seeking a test? I understand that the whole issue may be hard to investigate.	the common cold. This may have led to a rather low positivity rate amongst those showing symptoms. Our data does not allow a comparison between the information provided by an AEN in the presence and absence of symptoms, simply because asymptomatic persons without an AEN could not be detected (unless they presented with another test indication). We have clarified our interpretation of this result as follows: “In other words, the DPT system provided some information on whether COVID-19 was the cause of any symptoms.” (lines 271-272). Indeed, as the reason for testing was self-reported, it may also be biased. During the study period, unvaccinated individuals exposed to an infectious case were legally required to quarantine until they underwent a second test (clarified, line 543-545). However, with vaccination rates over 90% from September 2021 onwards (lines 433-434), we do not think this could have had a large influence. Also, persons unwilling to reveal an exposure for any reason could select a vague test indication (“concern about a possible infection”, see Supplementary Table 1). Individuals reporting this test indication were excluded from the infection risk analysis, as described.
Minor points: lines 76-77: I can understand the issue of the authors with the term "secondary attack rate", but this term is vaguely defined in the literature and used quite often in the same context as "infection risk" in this ms, even when the direction of transmission is not known.	“Secondary attack rate” is indeed vaguely defined in the literature. Many authors use the term for the infection risk amongst persons with certain characteristics. However, it is also used to describe the transmission risk from an infectious person to an exposed person, a measure which is not equivalent (e.g., https://doi.org/10.1371/journal.pcbi.1008601). To avoid confusion, we prefer the more specific term “infection risk”. We have added the reference above to support the choice of wording. (line 67)
Lines 96-98: "cannot be used" is not technically correct and it is far too strong. It is true that GAEN poses significant limits.	Revised as follows: “[...] the GAEN system [...] limits the study of transmission chains and tracking of certain performance indicators [...]” (line 87-88)
Line 103: to my memory, there is at least one study cited there (Wymant et al) that infers the impact of DCT based on statistical regression rather than modelling (and therefore does not require especially strong assumptions).	The study by Wymant et al indeed has a statistical regression component. We revised the sentence: “Although such studies can give an idea of the overall impact of DPT in specific contexts, they cannot compare DPT directly to MCT in terms of overlap in detected cases or timeliness.” (lines 94-96)

REVIEWERS' COMMENTS

Reviewer #2 (Remarks to the Author):

The authors have provided satisfactory answers to all my comments.

In my opinion, this manuscript can be published once some minor points are clarified:

- lines 223-233: one of the most impressive results of this manuscript is not the low effectiveness of DPT, but the very high effectiveness of MCT in this setup: 2/3 of all detected cases were found through MCT rather than symptoms. This number is quite exceptional if true, and my suspect is that there may be reporting biases at play, e.g. MCT could have been actively pushing students to get tested.

The author should comment on this, since it may significantly distort the MCT/DPT comparison and therefore the MCT success rate reported here. It shouldn't affect the absolute DPT success rate.

- line 277: as remarked earlier, this sentence is nonsensical since neither MCT nor DPT trace all contacts, and therefore the comparison has no absolute value. The authors should clarify that this comparison applies specifically to the context of the study, i.e. the university MCT criteria for tracing and the settings of the Belgian app.

- lines 308-310: "expanding MCT criteria (e.g., the close contact definition or the contact elicitation window)" is likely to include contacts with a much lower predictive value than the one reported for MCT, since it comprises lower-risk contacts compared to the current MCT criteria. Given the complex balance of pros/cons, I would suggest a more generic statement.

- lines 355-356: I already commented previously that Wymant et al presented also non-model-based evidence of effectiveness.

Second point-by-point response to reviewer comments

We would like to thank the reviewers again for their constructive feedback. The table below lists our response to each comment.

Reviewer #2 (Remarks to the Author)	Point-by-point response
The authors have provided satisfactory answers to all my comments. In my opinion, this manuscript can be published once some minor points are clarified:	
- lines 223-233: one of the most impressive results of this manuscript is not the low effectiveness of DPT, but the very high effectiveness of MCT in this setup: 2/3 of all detected cases were found through MCT rather than symptoms. This number is quite exceptional if true, and my suspect is that there may be reporting biases at play, e.g. MCT could have been actively pushing students to get tested. The author should comment on this, since it may significantly distort the MCT/DPT comparison and therefore the MCT success rate reported here. It shouldn't affect the absolute DPT success rate.	An additional limitation was described in the Discussion section (lines 410-414): “Fourth, despite a low number of manually traced contacts per case, the majority of cases was detected through MCT, rather than symptomatic screening. This could indicate an unusually effective MCT system or reporting bias, resulting in a distorted comparison with the epidemiological impact of DPT. However, this should not affect the observed absolute success rate of DPT.”
- line 277: as remarked earlier, this sentence is nonsensical since neither MCT nor DPT trace all contacts, and therefore the comparison has no absolute value. The authors should clarify that this comparison applies specifically to the context of the study, i.e. the university MCT criteria for tracing and the settings of the Belgian app.	As suggested, we specified that the comparison applies specifically to this context (lines 275-277): “Crucially, for contacts traced only digitally, the infection risk was significantly lower (risk ratio: 0.51, CI: 0.33-0.80) compared to manually traced non-app users, indicating a lower positive predictive value of DPT in the context of this study (Figure 2).”
- lines 308-310: "expanding MCT criteria (e.g., the close contact definition or the contact elicitation window)" is likely to include contacts with a much lower predictive value than the one reported for MCT, since it comprises lower-risk contacts compared to the current MCT criteria. Given the complex balance of pros/cons, I would suggest a more generic statement.	The statement was made more generic, by excluding the examples between brackets (lines 307-309): “If lowering the thresholds for DPT results in a higher quarantine burden per detected case compared to MCT, expanding MCT criteria may be the more efficient intervention.”
- lines 355-356: I already commented previously that Wymant et al presented also non-model-based evidence of effectiveness.	Wymant et al indeed presented convincing evidence that, as in our setting, digital proximity tracing through the NHS COVID-19 app had an epidemiological impact. They used a modelling approach and a statistical regression approach to quantify the absolute impact. However, it remains true that they could not compare DPT to symptomatic screening, compare the speed of

	digital notifications, or quantify overlap between case detection strategies. Their comparison of DPT to MCT effectiveness was limited to calculations with assumed figures in Supplementary Table 1. For example, the article compared the SAR (secondary attack rate) of DPT contacts to an assumed 6.9% for MCT, based on an interpretation of a document which actually reported an overall SAR of 12.7% (Variant of Concern Technical Briefing 3, available online). This has been clarified (lines 352-355): “Importantly, these studies focused on the absolute epidemiological impact of DPT. They could not make a direct comparison between cases identified by MCT, DPT, symptomatic screening, or a combination, without assumptions regarding the speed, comprehensiveness, and efficiency of different case detection strategies.”
--	---